# *Sporothrix brasiliensis* and Feline Sporotrichosis in the Metropolitan Region of Rio de Janeiro, Brazil (1998–2018)

**DOI:** 10.3390/jof8070749

**Published:** 2022-07-20

**Authors:** Jéssica Sepulveda Boechat, Manoel Marques Evangelista Oliveira, Isabella Dib Ferreira Gremião, Rodrigo Almeida-Paes, Ana Caroline de Sá Machado, Rosely Maria Zancopé-Oliveira, Raquel de Vasconcellos Carvalhaes Oliveira, Débora Salgado Morgado, Maria Lopes Corrêa, Anna Barreto Fernandes Figueiredo, Rodrigo Caldas Menezes, Sandro Antonio Pereira

**Affiliations:** 1Laboratory of Clinical Research on Dermatozoonoses in Domestic Animals, Evandro Chagas National Institute of Infectious Diseases, Oswaldo Cruz Foundation, Rio de Janeiro 21040-900, Brazil; isabella.dib@ini.fiocruz.br (I.D.F.G.); karol.vet1@gmail.com (A.C.d.S.M.); debora.morgado@ini.fiocruz.br (D.S.M.); maria.correa@ini.fiocruz.br (M.L.C.); anna.figueiredo@ini.fiocruz.br (A.B.F.F.); rodrigo.menezes@ini.fiocruz.br (R.C.M.); sandro.pereira@ini.fiocruz.br (S.A.P.); 2Laboratory of Taxonomy, Biochemistry and Bioprospecting of Fungi, Oswaldo Cruz Institute, Oswaldo Cruz Foundation, Rio de Janeiro 21040-900, Brazil; manoel.marques@ioc.fiocruz.br; 3Mycology Laboratory, Evandro Chagas National Institute of Infectious Diseases, Oswaldo Cruz Foundation, Rio de Janeiro 21040-900, Brazil; rodrigo.paes@ini.fiocruz.br (R.A.-P.); rosely.zancope@ini.fiocruz.br (R.M.Z.-O.); 4Laboratory of Clinical Epidemiology, Evandro Chagas National Institute of Infectious Diseases, Oswaldo Cruz Foundation, Rio de Janeiro 21040-900, Brazil; raquel.vasconcellos@ini.fiocruz.br

**Keywords:** *Sporothrix brasiliensis*, molecular characterization, clinical aspects, epidemiology, sporotrichosis, cats

## Abstract

Feline sporotrichosis is enzootic in different regions of Brazil, especially in Rio de Janeiro. This study compared the genotype profiles of *Sporothrix* sp. isolated from cats in Rio de Janeiro between 1998 and 2018 and evaluated their association with clinical and epidemiological characteristics. One hundred nineteen *Sporothrix* sp. isolates from a cohort of cats with sporotrichosis seen at INI/Fiocruz were included. Clinical and epidemiological data were obtained from the medical records of the animals. T3B PCR fingerprinting was used for molecular identification of the *Sporothrix* species. All isolates were characterized as *Sporothrix brasiliensis*, with the observation of low intraspecific variation in 31 isolates (31.3%). The interval between lesion onset and first medical visit at INI/Fiocruz, as well as treatment duration until clinical cure, was longer in cats from the first decade of the epizootic. In addition, the frequency of the variables “good general status” and “presence of lymphadenomegaly” was higher among cats whose strains did not exhibit intraspecific variation. So far, *S. brasiliensis* has been the only species identified in feline cases of sporotrichosis since the beginning of the epizootic in Rio de Janeiro at INI/Fiocruz.

## 1. Introduction

Sporotrichosis is a subacute to chronic infection caused by pathogenic species of the genus *Sporothrix* [1]. The disease affects humans and a variety of animals, with cats being the most affected species [2,3]. 

Until 2007, sporotrichosis had been attributed to a single causative agent, the thermally dimorphic fungus *Sporothrix schenckii* [4,5]. However, after 2007, genetic variability started to be observed among the isolates morphologically identified as *S. schenckii* [4]. This variability led to the proposal of eight pathogenic species, which include *S. schenckii sensu stricto*, *S. brasiliensis*, *S. globosa*, and *S.*
*luriei* that belong to the clinical clade, and *S. mexicana*, *S. pallida*, *S. chilensis*, and *S. humicola* that belong to the environmental clade [5,6,7,8,9,10,11]. To date, the genus *Sporothrix* comprises 53 species; however, *S. schenckii*, *S. brasiliensis*, and *S. globosa* are currently the main species of clinical interest for humans [1,9,10] and *S. brasiliensis* and *S. schenckii sensu stricto* for cats and dogs [12,13,14,15,16,17,18]. *Sporothrix brasiliensis* has been described as an emerging and highly pathogenic species for humans and cats, which is found in Brazil, Paraguay, and Argentina [5,12,16,19,20,21,22,23,24,25,26] 

Starting in 1998, a high frequency of sporotrichosis cases has been observed among humans, dogs, and cats in the metropolitan region of Rio de Janeiro, Brazil. This was the first epidemic of this mycosis in the form of a zoonosis described in the literature, whose main form of transmission was associated with the scratch and/or bite of sick cats [3,27,28]. Since then, the Evandro Chagas National Institute of Infectious Diseases (INI)/Oswaldo Cruz Foundation (Fiocruz), a leading referral center for medical mycology in Brazil, has diagnosed approximately 5000 human cases over the period from 1998 to 2015 and 5113 feline cases over the period from 1998 to 2018 [29]. 

Generally, cats with sporotrichosis initially present localized skin lesions that tend to progress to multiple skin lesions at different anatomical sites and fatal systemic involvement, associated or not with extracutaneous signs [30]. The definitive diagnosis of sporotrichosis requires isolation of the fungus in culture medium [31]. Since species of the genus *Sporothrix* are morphologically and physiologically similar, caution is necessary when only phenotypic characterization is considered [16,17,20,32,33,34,35,36] and molecular methods have been strongly recommended for species identification [16,20,35,36]. 

Despite clinical studies on feline sporotrichosis, the relationship between the clinical and epidemiological characteristics of cats and the identification of species and their genotypic variations within the genus *Sporothrix* are still poorly explored. Since the first characterization of isolates in animals from Rio de Janeiro in 2013 [12], few studies have characterized feline isolates in this region after the description of the new species in 2007 [16,17,22]. Species of the genus *Sporothrix*, as well as intraspecific variations in *S. brasiliensis* isolates obtained from humans, have been associated with different degrees of virulence [19,37]. Therefore, studies characterizing circulating isolates of *Sporothrix* sp. and evaluating their association with clinical and epidemiological factors are important to identify possible predictors that can assist in the treatment of feline sporotrichosis and in the prevention of transmission of this etiological agent [17]. Within this context, the objectives of the present study were to compare the genotype profiles of *Sporothrix* sp. isolated from cats in the metropolitan region of Rio de Janeiro during two periods between 1998 and 2018, and to evaluate their association with clinical and epidemiological characteristics. 

## 2. Materials and Methods

### 2.1. Isolates

This study included 119 viable isolates of *Sporothrix* sp. obtained from a cohort of cats with sporotrichosis seen at the Laboratory of Clinical Research on Dermatozoonoses in Domestic Animal (Lapclin-Dermzoo), INI/Fiocruz, Rio de Janeiro, Brazil, between 1998 and 2018. The *Sporothrix* sp. were divided into two groups: group A consisted of 20 isolates obtained over the period from 1998 to 2008 (the first 10 years of epidemic/endemic zoonotic sporotrichosis in Rio de Janeiro), kept at the Laboratory of Mycology (INI/Fiocruz), and group B consisted of 99 isolates obtained over the period from 2013 to 2018, kept at Lapclin-Dermzoo (INI/Fiocruz).

### 2.2. Clinical and Epidemiological Data

The clinical and epidemiological data were obtained by reviewing the medical records of the animals from which the isolates were obtained. The following clinical variables were analyzed: general health status, presence and distribution of skin lesions, presence of lymphadenomegaly, presence of respiratory signs, presence of mucosal lesions, and outcome.

The cats were classified into three groups according to the distribution of skin lesions: L1 (skin lesions at one site), L2 (skin lesions at two non-adjacent sites), and L3 (skin lesions at three or more non-adjacent sites) [30]. The following epidemiological variables were evaluated: municipality of residence, reproductive status (if the animal was neutered or not), sex, breed, and outdoor access of the cat. For the purpose of this study, the municipalities of residence of the cats were divided into three areas: Metropolitan region I (Rio de Janeiro, Belford Roxo, Duque de Caxias, Mesquita, Nova Iguaçu, Queimados and São João de Meriti), Metropolitan region II (São Gonçalo and Maricá), and Baixada Litorânea (Araruama and Cabo Frio) [38].

Licenses for therapeutic and diagnostic management, with consequent isolation of *Sporothrix* from the animals with sporotrichosis included in this study, were granted in studies approved by the Ethics Committee on Animal Use of Fiocruz (CEUA/Fiocruz).

### 2.3. Evaluation of Strain Viability and Molecular Analysis

For T3B PCR fingerprinting, the isolates were recovered and subcultured in PDA medium (Difco™; Becton, Dickinson and Company, Sparks, MD, USA) at 25 °C [20] in order to evaluate the macro- and microscopic viability of each isolate. 

Genomic DNA was extracted from filamentous colonies after 14 days of growth on PDA medium using chloroform:isoamyl alcohol (24:1) according to a previously described protocol [20]. DNA concentration was quantified in a NanoDrop™ spectrophotometer (GE Healthcare). All samples were diluted to a working concentration of 25 ng.

The universal T3B primer (5′-AGGTCGCGGGTTCGAATCC-3′) was used for the hybridization of genomic DNA in order to distinguish species of the genus *Sporothrix* and to evaluate genetic similarity [39,40]. DNA of the following reference strains was used as control: *S. brasiliensis* (IPEC16490), *S. globosa* (IPEC27135), *S. mexicana* (MUM11.02), and *S. schenckii* (IPEC27722). For the PCR assay, 2.5 μL 10X buffer with KCl, 0.5 μL of each dNTP (0.2 mM), 2 μL of 50 mM MgCl_2_, 0.2 μL Platinum Taq DNA polymerase, and 1 μL of T3B primer were mixed in a final volume of 21 μL for each sample. The hybridization conditions in the Veriti™ 96-Well Thermal Cycler (Applied Biosystems, Waltham, MA, USA) were: initial denaturation at 95 °C for 10 min, followed by 36 cycles of denaturation at 95 °C for 30 s, annealing at 52 °C for 30 s and extension for 1 min and 20 s at 72 °C, and a final extension at 72 °C for 10 min.

Amplification was confirmed by visualization of the amplicons after electrophoresis on 1.2% agarose gel (UltraPure Agarose, Invitrogen, Waltham, MA, USA) in 0.5X TBE buffer (0.1 M Tris, 0.09 M boric acid, 0.001 M EDTA, pH 8.4). Six microliters of the PCR product was added to each slot and the gel was run at 70 V for 90 min. The gel was then stained with ethidium bromide (0.5 µg/mL) for 30 min and washed with distilled water for an additional 30 min before examination under a UV transilluminator (Hoefer Scientific Inc., Holliston, MA, USA). This technique generates different bands for each species of the genus *Sporothrix*, thus permitting their differentiation [40]. The T3B PCR fingerprinting profiles of each group were analyzed using the Bionumerics 5.1 software (Applied Maths BVBA, Sint-Martens Latem, Belgium).

### 2.4. Statistical Analysis

For exploratory analysis of the data, the frequency distribution was calculated for categorical variables and summary measures (median, minimum, and maximum) were used for quantitative variables. Fisher’s exact test and Pearson’s chi-square test were used to assess the association between categorical variables. Differences in treatment duration and in the onset of lesions were compared between groups by the nonparametric Mann–Whitney test. A *p*-value < 0.05 indicates significant associations in the statistical tests. Due to the sample size, 95% confidence intervals for proportions were provided and 10-percentage point differences between groups are also indicated because of the criticism of the exclusive use of *p*-values in decision making [41]. All analyses were stratified by group and were performed using the R 3.6.1 software [42]. 

## 3. Results

In the present study, 119 isolates of *Sporothrix* sp. of cats with sporotrichosis presented cutaneous lesions associated or not with mucosal involvement (Figure 1 and Figure 2).

The median age of the 20 cats with sporotrichosis of group A (*Sporothrix* sp. isolates obtained between 1998 and 2008) was 24 months (7–72 months). Regarding their origin, all animals were from Metropolitan region I, with 13 (65.0%) from the municipality of Rio de Janeiro and the remaining from municipalities of the Baixada Fluminense, including Duque de Caxias (*n* = 3; 15.0%), São João de Meriti (*n* = 2; 10.0%), Nilópolis (*n* = 1; 5.0%), and Nova Iguaçu *(n* = 1; 5.0%). 

The median age of the 99 cats with sporotrichosis of group B (*Sporothrix* sp. isolates obtained between 2013 and 2018) was 24 months (2-96 months). Sixty-three animals (63.6%) were from different regions of the municipality of Rio de Janeiro and 26 (26.3%) from municipalities of the Baixada Fluminense, including Duque de Caxias (*n* = 7; 7.1%), Nova Iguaçu (*n* = 7; 7.1%), São João de Meriti (*n* = 4; 4.0%), Mesquita (*n* = 4; 4.0%), Belford Roxo (*n* = 3; 3.1%), and Queimados (*n* = 1; 1.0%), all of them belonging to Metropolitan region I. Seven cats (7.1%) were from the municipality of São Gonçalo and one (1.0%) from Maricá (Metropolitan region II). The two remaining animals were from Cabo Frio (*n* = 1; 1.0%) and Araruama (*n* = 1; 1.0%), municipalities of the Baixada Litorânea. 

Table 1 and Table 2 show the simple frequency distribution of the clinical and epidemiological characteristics of the 119 cats of both groups. It was not possible to evaluate the outdoor access of 11 cats of group A, since the data were not recorded.

In group A, the median interval between the onset of lesions and the first medical visit was 8 weeks (1–24 weeks). Five of the 20 cats achieved clinical cure as outcome, with a median treatment duration of 40 weeks (20–84 weeks). In group B, the median interval between the onset of lesions and the beginning of treatment was 4 weeks (1–24 weeks). Thirty-two cats achieved clinical cure and the median treatment duration was 17 weeks (4–56 weeks). The interval between lesion onset and first medical visit (*p* = 0.029, Mann–Whitney test) was significantly longer in group A compared to group B, as was the treatment duration until cure (*p* = 0.012, Mann–Whitney test). No other clinical or epidemiological differences were observed between the two groups. 

All isolates included in the study were characterized as *S. brasiliensis*. Phylogenetic analysis of group A showed that all isolates were aligned to the reference strain (IPEC 16490) and no intraspecific genotypic variation was observed. In group B, phylogenetic analysis identified low intraspecific variation in 31 isolates (31.3%) when compared to the *S. brasiliensis* reference strain (Appendix A). 

Table 3 and Table 4 show the association with the clinical and epidemiological variables of cats according to the presence or absence of intraspecific genotypic variation in the *S. brasiliensis* isolates. The frequency of a good general status (*p* = 0.034, Pearson’s chi-square test) and lymphadenomegaly (*p* = 0.006, Pearson’s chi-square test) was higher among cats with isolates without intraspecific variation compared to those with isolates exhibiting intraspecific genotypic variation in both groups (A and B).

Regarding the spatial distribution of the 31 isolates that exhibited intraspecific variation, 24 (77.4%) were from Metropolitan region I, five (16.1%) from Metropolitan region II, and two (6.5%) from the Baixada Litorânea. Among the remaining 88 isolates without intraspecific variation, 85 (96.6%) were isolates from cats living in Metropolitan region I and three (3.4%) from animals living in Metropolitan region II (Figure 3).

## 4. Discussion

Since the description of new species of the genus *Sporothrix* in 2007, studies have been conducted in sporotrichosis-endemic areas in order to identify clinical isolates of humans and animals. However, worldwide, there are few animal isolates of *Sporothrix* sp. that have been identified to the species level and published over the last 12 years [11,12,13,14,16,17,18,21,22,23,43,44,45,46,47,48,49,50,51]. In the present study, we molecularly characterized 119 *Sporothrix* sp. isolates from cats diagnosed at INI/Fiocruz, Rio de Janeiro, Brazil, during two different periods corresponding to two decades of epizootic sporotrichosis (1998 and 2018). Our study is one of the largest samples of animal isolates of *Sporothrix* sp. identified in a single study in Rio de Janeiro state. Additionally, the clinical and epidemiological characteristics of cats were analyzed, and the fungal species of each isolate was characterized.

Cat-related zoonotic sporotrichosis has been widely reported in Brazil over the last two decades, especially in the state of Rio de Janeiro where it became hyperendemic [3,52]. However, cases have also been described in other countries such as the United States [53,54,55,56], Mexico [57], Panamá [58], Argentina [23], Paraguay [59], India [60], Malaysia [61,62], and China [63]. Molecular epidemiology studies are therefore important for understanding the profile of species that circulate in Rio de Janeiro, a region with the world’s largest number of animal sporotrichosis cases to date. In addition, it is important to know the profile of species circulating in new areas where feline sporotrichosis occurs.

Identification by molecular techniques is still costly and requires human resources specialized in molecular biology, which are not part of the diagnostic routine of mycology laboratories in Brazil. Consequently, the number of characterized feline isolates is still small [12,16] when compared to the number of diagnosed cats in the metropolitan region of Rio de Janeiro. Furthermore, feline isolates from the early period of the epizootic in the metropolitan region of Rio de Janeiro have not yet been characterized to the species level and few studies that identified animal isolates of the genus *Sporothrix* have associated them with clinical and epidemiological data [16,17]. 

The clinical profile of the feline population affected by sporotrichosis in Brazil is similar to that described in other countries such as the United States, Malaysia, and Argentina [13,14,25,64,65]. Regarding the epidemiological profile of cats with sporotrichosis, the findings of the present study agree with those described in previous studies conducted in Rio de Janeiro and in other states of Brazil [22,30,66,67,68,69,70] (a predominance of male, young adult, non-neutered cats with outdoor access and mongrel). In contrast, the authors of a study conducted in the United States reported that most cats with sporotrichosis are neutered breed animals [65]. This reproductive status may be explained by the fact that the concept of responsible guardianship is more widespread and applied in that country. An epidemiological profile of cats with sporotrichosis similar to that observed here has been reported in Argentina [25].

Other studies investigating feline isolates from Rio de Janeiro also identified only *S. brasiliensis*, in agreement with our findings [12,16,17,22]. These results support the hypothesis that *S. brasiliensis* has been the predominant species among cats in the metropolitan region of Rio de Janeiro since the beginning of the epizootic in 1998. However, other studies investigating human, feline, and canine isolates of *Sporothrix* sp. from Brazil have identified other species, including *S. schenckii sensu stricto* in a canine case [71], in feline cases [43,72], and in humans [20,34], as well as *S. luriei* in a canine case [43], and *S. globosa* only in humans [73]. The description of other *Sporothrix* species from Brazil, including in dogs and humans from the same geographical region of this study, indicates that they may be circulating among cats in Rio de Janeiro; however, these other *Sporothrix* species in cats from this region have not yet been confirmed.

Comparison of the clinical and epidemiological characteristics of cats of the two groups showed that, despite the high frequency of cats in good general health, 40% of the animals of group A had a regular/poor general status compared to 15.1% in group B. This difference might be explained by the fact that in the first 10 years of the epizootic (group A), the time to the first appointment was twice that observed in group B (2013–2018), probably because of the initial lack of knowledge of owners about the disease. Therefore, these cats arrived at the first medical visit with the disease at a more advanced stage and treatment duration was consequently longer when compared to group B. However, overall comparison of the clinical and epidemiological profile of cats of groups A and B showed similar characteristics of the animals. Furthermore, the clinical and epidemiological data described in other studies on cats diagnosed within the first 10 years of the epizootic were similar to the data of cats of group A included in the present study [30,66], suggesting that *S. brasiliensis* is the most prevalent species of feline sporotrichosis since the beginning of the epizootic.

There were no significant differences in the clinical–epidemiological profile or molecular profile of the *S. brasiliensis* isolates between the different regions of the state of Rio de Janeiro. However, studies involving a larger number of isolates are necessary to determine whether other pathogenic *Sporothrix* species circulate among cats in this region.

Small intraspecific variation was only detected in the *S. brasiliensis* isolates obtained between 2013 and 2018 (group B). The frequency of this intraspecific variation was similar to the frequency of 29.8% reported previously in a study including feline isolates from Rio de Janeiro obtained after the first decade of the epidemic between 2010 and 2011 [16]. Low intraspecific variation has also been observed in human clinical isolates of *S*. *brasiliensis* [37,39] and feline isolates from São Paulo [45]. One of the first studies in Brazil that characterized clinical *Sporothrix* sp. isolated from cats also reported the occurrence of intraspecific variation among isolates [12]. However, this variation was not detected in the 15 *S. brasiliensis* isolates from the metropolitan region of Rio de Janeiro included in this previous study [12]. The results of these investigations and of the present study suggest low genetic diversity of the fungus *S. brasiliensis* and that it is a clonal species. However, different genotypes of this species were found among feline isolates from Rio Grande do Sul, which differ from the isolates found in southeastern states such as Minas Gerais and São Paulo [12,45]. In addition, two different *S. brasiliensis* genotypes isolated from the same human patient in the metropolitan region of Rio de Janeiro who had cutaneous sporotrichosis transmitted by a cat bite exhibited different virulence profiles [37]. 

No intraspecific variation was observed in the isolates collected over the period from 1998 to 2008 (group A). In contrast, Reis et al. [74], analyzing feline isolates of *Sporothrix* sp. at the beginning of the epidemic, between 1998 to 2001, found a frequency of genetic variability among isolates of about 19%; however, the authors used different methodologies (PCR-RAPD and M13 PCR-fingerprinting). In the present study, an association was observed between isolates without intraspecific variation and the clinical variables “good general status” and “presence of lymphadenomegaly”. The response of the organism to fungal infection can influence the occurrence of lymphadenomegaly [75] and consequently the animal’s general health status. The causes for the occurrence of these intraspecific variations in feline isolates of *S. brasiliensis* are still unknown and further studies are therefore necessary to better understand the underlying mechanisms. 

Regarding possible differences in clinical-therapeutic management between the periods of the isolates studied, a higher frequency of the use of potassium iodide for the treatment of cats with sporotrichosis was observed in the first years of the 2010 decade in Rio de Janeiro. Even if the cats included in this study were not undergoing treatment at the time of material collection for DNA extraction, the fungus transmitted to the animal may have originated from an animal that had been treated with potassium iodide. It is known that iodide exerts a direct effect on the yeast phase of *Sporothrix* sp. [76]. We therefore suggest investigating in the future the possible association between some mutations in *S. brasiliensis* isolates and the use of this drug.

Identification of feline isolates of *Sporothrix* sp. using molecular techniques was not performed at the beginning of the epizootic. Unfortunately, one limitation of the present study is that, at the beginning of the historical series of feline sporotrichosis in Rio de Janeiro, no support infrastructure was available for the storage and identification of *Sporothrix* sp. strains, mainly because the magnitude of this public health problem was unknown at that time. This fact highlights the importance of the present findings.

It is important to continue molecular epidemiology studies in order to identify new species of the genus *Sporothrix* that could be circulating among animals, since species other than *S. brasiliensis* circulate among humans and dogs living in the same region. T3B PCR fingerprinting is a simple, fast, inexpensive, and reliable technique for the characterization to species level; however, the method has limitations, especially for elucidating and better understanding the intraspecific variations found in this study. We therefore suggest that future studies employ other molecular methods such as whole-genome sequencing, expanding the knowledge about animal isolates of *Sporothrix* sp. that circulate in enzootic areas. This greater genetic knowledge about isolates of the genus *Sporothrix* would help to develop effective strategies for the prevention and control of sporotrichosis.

*Sporothrix brasiliensis* has been the only species identified in feline cases of sporotrichosis since the beginning of the epizootic in Rio de Janeiro. The isolates of this species of fungus show low intraspecific variation, which is reflected in low variation in the clinical characteristics of cats over two decades of sporotrichosis in Rio de Janeiro.

## Figures and Tables

**Figure 1 jof-08-00749-f001:**
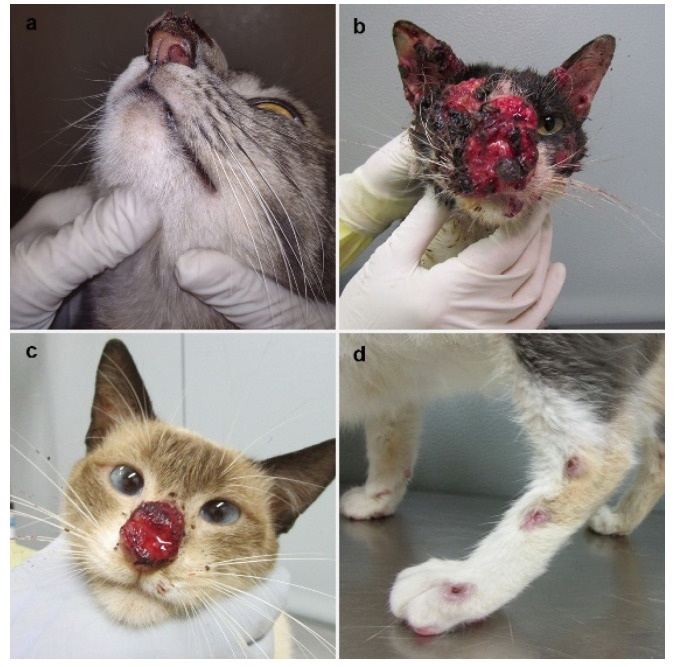
Clinical presentations of feline sporotrichosis caused by *Sporothrix brasiliensis*. (**a**) Swelling of the nasal bridge with crusted skin ulcer and nasal mucosa lesion causing narrowing of the left nostril. (**b**) Multiple skin lesions partially covered by hematic crusts draining serosanguinous exudate on the cephalic region with the involvement of nasal/ocular mucosa. (**c**) Crusted skin ulcer on the nasal bridge and nasal planum, draining serosanguinous exudate. (**d**) Ascending nodular lymphangitis on the left hindlimb.

**Figure 2 jof-08-00749-f002:**
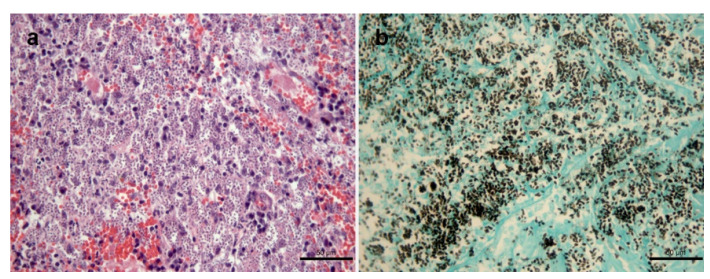
Histological changes in the skin lesion of a cat with sporotrichosis caused by *Sporothrix brasiliensis*. (**a**) Granulomatous dermatitis exhibiting abundant yeasts inside macrophages, lymphocytes and plasma cells and vascular congestion. Hematoxylin and eosin. (**b**) Abundant black-stained round or cigar-shaped yeasts. Grocott’s methenamine silver stain.

**Figure 3 jof-08-00749-f003:**
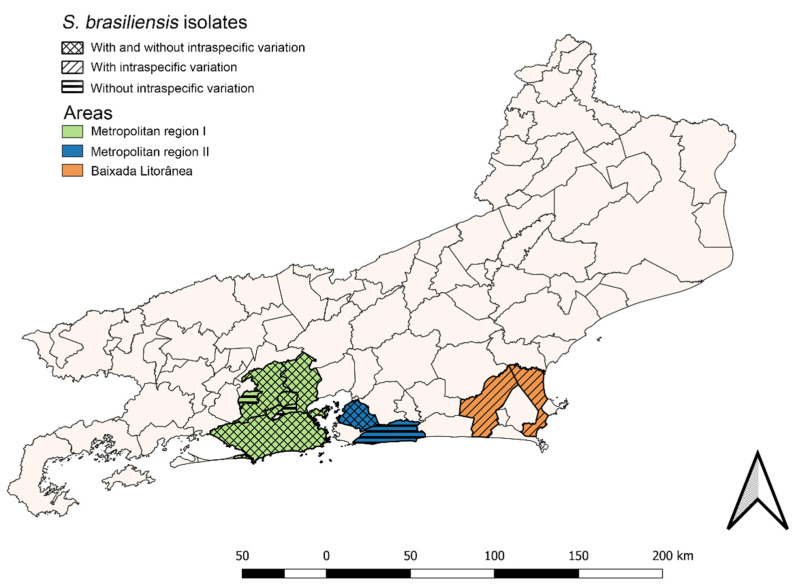
Geographic distribution of the 119 feline cases of sporotrichosis seen at the Laboratory of Clinical Research on Dermatozoonoses in Domestic Animals (INI/Fiocruz), Rio de Janeiro, Brazil (1998 to 2018).

**Table 1 jof-08-00749-t001:** Distribution of epidemiological variables of the 119 cats from which the *Sporothrix* sp. isolates were obtained between 1998 and 2018 (groups A and B), Rio de Janeiro, Brazil.

Variables		Group A			Group B		
		*n*	%	95% CI *	*n*	%	95% CI *
**Neutering**	Yes	2	10.0	1.7–33.1	46	46.5	36.4–56.7
	No	18	90.0	66.8–98.2	53	53.5	43.2–63.5
**Breed**	Mongrel	18	90.0	66.8–98.2	91	91.9	84.2–96.1
	Siamese	2	10.0	1.7–33.1	8	8.1	3.8–15.7
**Outdoor access**	Yes	8	88.9	19.9–63.5	80	80.8	71.4–87.7
	No	1	11.1	0.2–26.9	19	19.2	12.2–28.6
**Sex**	Male	13	65.0	40.9–83.6	83	83.8	74.7–90.2
	Female	7	35.0	16.3–59.1	16	16.2	9.7–25.2

* CI: confidence interval.

**Table 2 jof-08-00749-t002:** Distribution of clinical variables of the 119 cats from which the *Sporothrix* sp. isolates were obtained between 1998 and 2018 (groups A and B), Rio de Janeiro, Brazil.

Variables		Group A			Group B		
		*n*	%	95% CI *	*n*	%	95% CI *
**General health status**	Good	12	60.0	36.4–80.0	84	84.9	75.9–90.9
	Regular/poor	8	40.0	19.9–63.5	15	15.1	9.0–24.1
**Skin lesions**	Present	20	100.0	-	98	99.0	93.6–99.9
	Absent	0	-	-	1	1.0	0.05–6.30
**Distribution of skin lesions ****	L1	4	20.0	6.6–44.2	28	28.3	19.9–38.3
	L2	4	20.0	6.6–44.2	17	17.2	10.6–26.3
	L3	12	60.0	36.4–80.0	54	54.5	44.2–64.4
**Lymphadenomegaly**	Present	13	65.0	40.9–83.6	68	68.7	58.4–77.4
	Absent	7	35.0	16.3–59.1	31	31.3	22.5–41.5
**Respiratory signs**	Present	10	50.0	29.9–70.1	46	46.5	36.4–56.7
	Absent	10	50.0	29.9–70.1	53	53.5	43.2–63.5
**Mucosal lesions**	Present	11	55.0	32.0–76.1	36	36.3	27.1–46.6
	Absent	9	45.0	23.8–67.9	63	63.4	53.3–72.8
**Outcome**	Favorable (cure)	5	25.0	9.5–49.4	33	33.3	24.3–43.6
	Unfavorable	15	75.0	50.5–90.4	66	66.7	56.3–75.6

* CI: confidence interval. ** L1 (skin lesions at one site), L2 (skin lesions at two non-adjacent sites), and L3 (skin lesions at three or more non-adjacent sites).

**Table 3 jof-08-00749-t003:** Distribution of epidemiological variables in the 119 cats with sporotrichosis according to the presence or absence of intraspecific genotypic variation in the *S. brasiliensis* isolates obtained from these animals. Rio de Janeiro, Brazil (1998 to 2018).

Variables		With Intraspecific Variation (*n* = 31)	95% CI *	Without Intraspecific Variation (*n* = 88)	95% CI *	*p*-Value
**Neutering**	Yes	13 (41.9%)	25.1–60.7	35 (33.8%)	29.6–50.7	0.832
	No	18 (58.1%)	39.2–74.9	53 (66.2%)	49.2–70.3	
**Breed**	Mongrel	28 (90.3%)	73.0–97.4	81 (92.0%)	83.7–96.4	0.766
	Siamese	3 (9.7%)	2.5–26.9	7 (8.0%)	3.5–16.2	
**Outdoor access**	Yes	22 (71.0%)	51.7–85.1	66 (75.0%)	64.4–83.3	0.074
	No	9 (29.0%)	14.8–48.2	11 (12.5%)	6.7–21.6	
**Sex**	Female	5 (16.1%)	6.1–34.4	18 (20.4%)	12.8–30.6	0.599
	Male	26 (83.9%)	65.5–93.9	70 (79.6%)	69.3–87.1	

* CI: confidence interval.

**Table 4 jof-08-00749-t004:** Distribution of clinical variables in the 119 cats with sporotrichosis according to the presence or absence of intraspecific genotypic variation in the *S. brasiliensis* isolates obtained from these animals. Rio de Janeiro, Brazil (1998 to 2018).

Variables		With Intraspecific Variation (*n* = 31)	95% CI *	Without Intraspecific Variation (*n* = 88)	95% CI *	*p*-Value
**General health status**	Good	29 (93.5%)	77.1–98.8	67 (76.1%)	65.6–84.3	0.034 **
	Regular/Poor	2 (6.5%)	1.1–22.8	21 (23.9%)	15.6–34.3	
**Distribution of skin lesions ******	L1	7 (22.6%)	10.2–41.5	25 (28.4%)	19.5–39.1	0.814
	L2	6 (19.3%)	8.1–38.0	15 (17.1%)	10.1–26.8	
	L3	18 (58.1%)	25.1–60.7	48 (54.5%)	43.6–65.0	
**Respiratory signs**	Present	13 (41.9%)	39.2–74.9	43 (48.9%)	38.1–59.6	0.506
	Absent	18 (58.1%)	39.2–74.9	45 (51.1%)	40.3–61.8	
**Skin lesions**	Present	31 (100%)	***	87 (98.9%)	92.9–99.9	***
	Absent	0	***	1 (1.1%)	0.5–7.0	
**Mucosal lesions**	Present	10 (32.2%)	17.3–51.4	37 (42.0%)	31.7–53.0	0.337
	Absent	21 (67.8%)	48.5–82.6	51 (58.0%)	46.9–68.2	
**Lymphadenomegaly**	Present	15 (48.3%)	30.5–66.6	66 (75.0%)	64.4–83.3	0.006 **
	Absent	16 (51.7%)	33.3–69.4	22 (25.0%)	16.6–35.5	
**Outcome**	Favorable (cure)	11 (35.5%)	19.8–54.6	27 (30.7%)	21.5–41.5	0.621
	Unfavorable	20 (64.5%)	45.3–80.1	61 (69.3%)	58.4–78.4	

* CI: confidence interval. ** *p* < 0.05, Fisher’s exact test. *** It was not possible to calculate the *p*-value or CI because of the small number in the category. **** L1 (skin lesions at one site), L2 (skin lesions at two non-adjacent sites), and L3 (skin lesions at three or more non-adjacent sites).

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
