# Peer review of "Sporothrix brasiliensis and Feline Sporotrichosis in the Metropolitan Region of Rio de Janeiro, Brazil (1998–2018)"

_jof, 2022, doi:10.3390/jof8070749_

Round 1

Reviewer 1 Report

This is a fine study about Sporothrix brasiliensis infecting cats of the metropolitan area of Rio de Janeiro. The study revealed that all cats were infected by Sporothrix brasiliensis which is of considerable epidemiologic importance.

The article is well written and concise, However, I would have liked a few clinical and histopathological figures.

Author Response

Dear reviewer,

On behalf of all the authors, I would like to thank you for taking the time to consider our manuscript and for the suggestions to improve it.

Your sincerely,

Jéssica Boechat

Point 1: The article is well written and concise, However, I would have liked a few clinical and histopathological figures.

Response 1: The clinical and histopathological figures and their captions were included for improvement the manuscript. Figure 1 - page 4, lines 148-152. Figure 2, page 4, lines 154-157

Reviewer 2 Report

The manuscript of Boechat et al., entitled "Sporothrix brasiliensis and feline sporotrichosis in the metropolitan region of Rio de Janeiro, Brazil (1998-2018)" reports a genotypic characterization of Sporothrix isolates from cats in Rio de Janeiro between 1998 and 2018, by the use of T3B PCR fingerprinting method, followed by statistical association analysis with clinical and epidemiological characteristics.
The manuscript is well-presented and, in my opinion, it tackles an important aspect related to sporotrichosis in Brazil.
The experimental methodology used is appropriate for the intended objectives and in general, the paper is scientifically sound.

I only have a few minor comments (see below).

1) Pag. 1 line 41: "The genus Sporothrix comprises approximately 53 species.". 53 seems to be quite precise, so authors should remove "approximately" and add "To date, the genus Sporothrix comprises 53 species" for clarity.

2) Page 4 lines 163 and 166: please carefully check the date stated in "obtained between 1998 and 2008 (groups A and B)". Group B wasn't referred to the period 2013-2018?

3) Table 1. line Outdoor access: for group A only 8 cats had outdoor access, while 1 did not. What about the remaining 11 cats? If the data was not recorded, please clearly state it in the capital or correct the table.

4) Pag. 5 lines 180-184: Authors stated that "group A showed no intraspecific genotypic variation was observed" while for the group B "low intraspecific variation in 31 isolates" was found, but there are no electrophoretic bands shown in the manuscript. In my opinion, it's advisable to add, as supplementary material, a figure showing all genotypes identified in this study, in order to give the appropriate support to the described results, as was done in Figure 2 of the reference [40].

5) Pag. 7 lines 216-218: Please add references to this statement.

6) Pag. 8 lines 247-257: Please correct the scientific names in italics.

Author Response

Dear Reviewer,

The authors would like to thank you for taking the time to consider our manuscript and for all the comments and suggestions to improve it. We expect to have properly complied with yours remarks.

Your sincerely,

Jéssica Boechat

I only have a few minor comments (see below).

Point 1: Pag. 1 line 41: "The genus Sporothrix comprises approximately 53 species.". 53 seems to be quite precise, so authors should remove "approximately" and add "To date, the genus Sporothrix comprises 53 species" for clarity.

Response 1: The sentence was modified in order to clarify and the word “approximately” was removed as suggested

Point 2: Page 4 lines 163 and 166: please carefully check the date stated in "obtained between 1998 and 2008 (groups A and B)". Group B wasn't referred to the period 2013-2018?

Response 2: The correction was made and the date “1998 and 2008” was substitute to “1998 and 2018”. Page 5, lines 178 and 181.

Point 3:  Table 1. line Outdoor access: for group A only 8 cats had outdoor access, while 1 did not. What about the remaining 11 cats? If the data was not recorded, please clearly state it in the capital or correct the table.

Response 3: We included the text: “It was not possible to evaluate the outdoor access of 11 cats of group A, since the data were not recorded.” Page 5, lines 175 and 176.

Point 4: Pag. 5 lines 180-184: Authors stated that "group A showed no intraspecific genotypic variation was observed" while for the group B "low intraspecific variation in 31 isolates" was found, but there are no electrophoretic bands shown in the manuscript. In my opinion, it's advisable to add, as supplementary material, a figure showing all genotypes identified in this study, in order to give the appropriate support to the described results, as was done in Figure 2 of the reference [40].

Response 4: We included two figures as supplementary material for better comprehension. The figures and their captions were included in the end of the revised version of the manuscript. In the phylogenetic tree, it's possible to observe the species identification and the intraspecific variation of the isolates, which allowed us to separate them into two groups, group 1 (without intraspecific variation) and group 2 (with intraspecific variation). Figure S1 - page 6, line 199.

The second figure add as supplementary material is a representative PCR fingerprinting profiles obtained with primer T3B for Sporothrix brasiliensis isolates, showing the differences in eletrophoretic bands profile in both groups. Figure S2 - page 6, line 199.

Point 5: Pag. 7 lines 216-218: Please add references to this statement.

Response 5:  In this statement we are referring to our study, so we changed the word "this" to "our study" for better comprehension. Page 7, lines 231 and 232.

Point 6: Pag. 8 lines 247-257: Please correct the scientific names in italics.

Response 6: The scientific names was corrected not only on page 8, but also on page 1, lines 15-26 and page 8, lines 249-271.

Reviewer 3 Report

There are reports of epidemiological studies of cat sprotrichosis in Brazil. Since this report does not go beyond the scope of the previous reports, I have determined that it is not novel.

Author Response

Point 1: There are reports of epidemiological studies of cat sprotrichosis in Brazil. Since this report does not go beyond the scope of the previous reports, I have determined that it is not novel.

Response 1: In our study we reported one of the largest samples of animal isolates of Sporothrix sp. identified in a single study in Rio de Janeiro state. Additionally, the clinical and epidemiological characteristics of cats were analyzed and the fungal species of each isolate was characterized. Other studies with feline, canine and human isolates from Brazil have already been carried out and it was described other species, besides S. brasiliensis, circulating in the country. The identification of Sporothrix sp. species is very important, because there is different virulence between the species of this fungus and S. brasiliensis is considered the most virulent.

The metropolitan region of Rio de Janeiro is considered a hyperendemic area of sporotrichosis associated with zoonotic transmission of the disease, starting in 1998. Our work was the first to carry out the characterization of isolates from the beginning of the sporotrichosis epizootic in Rio de Janeiro. In addition to being the first to compare the genotypic profiles of isolates from the beginning of the epizootic to the profile of isolates obtained nowadays. In order to seek possible changes in the profile of these animals over the years, we also carried out the association of the genotypic profile of the oldest isolates with the clinical and epidemiological characteristics of cats.

It is important to highlight again that S. brasiliensis was the only species identified in feline cases of sporotrichosis since the beginning of the epizootic in Rio de Janeiro. The isolates of this species of fungus show low intraspecific variation, which is reflected in the low variation in the clinical characteristics of cats over two decades of sporotrichosis in Rio de Janeiro. Epidemiological and molecular studies are extremely important, especially in epidemic areas, this way new species can be identified and the genotypic profile of isolates circulating in each region can be evaluated. This greater genetic knowledge about isolates of the genus Sporothrix would help to develop effective strategies for the prevention and control of sporotrichosis.